# Selective Breeding for Disease-Resistant *PRNP* Variants to Manage Chronic Wasting Disease in Farmed Whitetail Deer

**DOI:** 10.3390/genes12091396

**Published:** 2021-09-10

**Authors:** Nicholas Haley, Rozalyn Donner, Kahla Merrett, Matthew Miller, Kristen Senior

**Affiliations:** Department of Microbiology and Immunology, College of Graduate Studies, Midwestern University, Glendale, AZ 85331, USA; rdonner11@midwestern.edu (R.D.); Kmerrett35@midwestern.edu (K.M.); mmiller57@midwestern.edu (M.M.); ksenio@midwestern.edu (K.S.)

**Keywords:** prion, deer, *PRNP*, selective breeding, resistance, susceptibility, CWD

## Abstract

Chronic wasting disease (CWD) is a fatal transmissible spongiform encephalopathy (TSE) of cervids caused by a misfolded variant of the normal cellular prion protein, and it is closely related to sheep scrapie. Variations in a host’s prion gene, *PRNP*, and its primary protein structure dramatically affect susceptibility to specific prion disorders, and breeding for *PRNP* variants that prevent scrapie infection has led to steep declines in the disease in North American and European sheep. While resistant alleles have been identified in cervids, a *PRNP* variant that completely prevents CWD has not yet been identified. Thus, control of the disease in farmed herds traditionally relies on quarantine and depopulation. In CWD-endemic areas, depopulation of private herds becomes challenging to justify, leading to opportunities to manage the disease in situ. We developed a selective breeding program for farmed white-tailed deer in a high-prevalence CWD-endemic area which focused on reducing frequencies of highly susceptible *PRNP* variants and introducing animals with less susceptible variants. With the use of newly developed primers, we found that breeding followed predictable Mendelian inheritance, and early data support our project’s utility in reducing CWD prevalence. This project represents a novel approach to CWD management, with future efforts building on these findings.

## 1. Introduction

Chronic wasting disease (CWD) is a progressive and ultimately fatal transmissible spongiform encephalopathy (TSE) affecting both farmed and free-ranging populations of deer, elk, and other cervid species [1,2,3]. First reported in northern Colorado and southern Wyoming over five decades ago, CWD has since spread across much of North America and has recently been reported in free-ranging cervids in Scandinavia [4,5,6]. The causative agent of TSEs, a disease category that includes sheep scrapie, bovine spongiform encephalopathy, and human kuru and Creutzfeldt–Jakob disease, is a misfolded, protease-resistant prion protein often denoted PrP^res^ [7,8]. This misfolded prion protein is a conformer of a normal cellular prion protein, PrP^C^, a ~250 amino acid protein encoded by a single-copy gene (*PRNP*) that has been reported in a broad range of mammalian, avian, reptilian, amphibian, and piscine species [8,9]. Despite the wide prevalence of the *PRNP* gene and PrP^C^ in animal species, TSEs have so far only been reported in humans and ungulates, occasionally spilling over into carnivores [10,11,12,13,14,15,16,17,18].

Because of the direct link between PrP^C^ and TSEs, a host’s *PRNP* genotype and the primary structure of the PrP^C^ protein(s) encoded have been shown to contribute to varying degrees of susceptibility to species-specific prion diseases, in some cases completely preventing infection [19,20,21]. Dozens if not hundreds of *PRNP* alleles have been reported in sheep, for example, and prevalence studies have identified variants with polymorphisms at positions 136, 154, and 171 of the ovine prion gene that are associated with high levels of resistance to sheep scrapie [20]. Sheep that are homozygous for alanine at position 136 and arginine at positions 154 and 171 (e.g., A^136^R^154^R^171^/A^136^R^154^R^171^ or simply ARR/ARR) are thought to be completely resistant to the infectious, classical form of scrapie. Agricultural agencies in the United States and Europe incorporated this information into their sheep flock improvement programs, yielding sharp declines in cases of classical scrapie over the past two decades [22,23]. Similar variations in the *PRNP* gene have been identified in humans [24], with polymorphisms at positions 127 and 129 associated with resistance to kuru, a prion disease linked to cannibalistic practices of the Fore people of Papua New Guinea [25], and variant Creutzfeldt–Jakob disease (vCJD), linked to the consumption of beef contaminated with bovine spongiform encephalopathy (BSE) prions [26].

In cervids, species-specific polymorphisms have likewise been identified which correlate with CWD prevalence and disease progression, although none so far have been found to completely prevent infection as reported with sheep and scrapie. In North American elk (*Cervus canadensis*), a 132M→L polymorphism has been linked to reduced CWD susceptibility and a delayed onset of disease [27,28], while mule deer (*Odocoileus hemionus*) have a polymorphism at codon 225 (225S→F) that is linked to reduced disease prevalence, protracted disease course, and variations in disease pathology [29,30,31]. White-tailed deer (*Odocoileus virginianus*), the focus of the present study, have several *PRNP* variants that modulate susceptibility, including those commonly referred to as 95H, 96G, 96S, 116G, and 226K based on nonsynonymous single-nucleotide polymorphisms at *PRNP* positions 285 (a→c; 95Q→95H), 286 (g→a; 96G→96S), 347 (c→g; 116A→116G), and 676 (g→c; 226Q→226K) [32,33]. To date, these polymorphisms have been found to be mutually exclusive, which is to say that no haplotype has been found with polymorphisms at both 285 and 286, for example. Deer carrying a 96G variant (i.e., those without nonsynonymous SNPs at other positions, sometimes reported as Q*^95^*G*^96^*A*^116^*Q*^226^* or simply QGAQ), especially in a homozygous state (e.g., 96G/96G or 96GG), are highly susceptible to CWD, while deer carrying other *PRNP* variants, even heterozygous with 96G, show lower degrees of susceptibility and protracted disease courses [32,34,35,36,37,38,39,40]. Because some variants are incredibly rare in wild and farmed populations of whitetail deer, little is known about CWD susceptibility, pathogenesis, and transmission in, e.g., 95HH, 95H/226K, or 226KK animals.

Managing CWD in farmed cervids demands heightened surveillance to quickly identify infected animals, in both CWD-endemic areas where disease may spill over from wild deer herds, and CWD-free areas where the reverse may be true. Historically, early identification of CWD-positive deer in a private herd would result in quarantine of the property, subsequent depopulation of the premises, and, when available, indemnification of the herd owner who for all intents and purposes has lost a livelihood that was years or perhaps generations in the making [41]. In areas where CWD has become well established in free ranging cervids, depopulation of farmed herds has become increasingly difficult to justify, and early identification of positive animals in these herds may more practically lend itself to rapid intervention and management [28]. These intervention and management strategies may subsequently benefit and inform wild cervid management directives in these areas.

The present study arose from a timely collaboration with a whitetail deer farm and hunting preserve located in an area with longstanding reports of CWD in surrounding wild populations. At the start of the study, the farm had recently identified cases of the disease on one of three hunting preserves, although CWD was soon found on a second preserve. A third hunting property and two breeding sites remained unaffected. The goals of our study were to (1) determine the *PRNP* genotypes of breeding animals and work hand in hand with the farm owner to develop a breeding program to reduce the frequency of highly susceptible *PRNP* genotypes, (2) evaluate *PRNP* genotypes and associated CWD prevalence on the two CWD-positive hunting preserves prior to our planned interventions, and (3) assess the preliminarily effectiveness of selective breeding on reducing CWD prevalence on these two sites, while preventing the incursion of CWD on the remaining properties. On the first hunting preserve, our strategy was a “clean slate” approach, with the property depopulated over the initial phase of the project and subsequently repopulated with small numbers of animals with known genotypes for 12–16 months prior to harvesting. For the second and third premises, we initially proposed a genotypic shift in situ, with older, more susceptible animals gradually harvested over time and replaced with younger stock with known *PRNP* genotypes linked to reduced susceptibility. Over the course of the first phase of the study, our plans for the second and third sites shifted to parallel our approach on hunting site 1 based on increasing rates of prevalence on site 2. We first hypothesized that selective breeding would yield outcomes with predictable patterns of Mendelian inheritance, e.g., no specific *PRNP* variant would prove lethal in utero. Secondly, we hypothesized that, prior to intervention, CWD prevalence on the two hunting preserves would gradually increase over time due to high frequencies of the 96G variant and 96GG genotypes. Lastly, we hypothesized that genetic shifts toward animals with reduced susceptibility to CWD would result in lower disease prevalence on both CWD-positive hunting preserves.

In this first phase of a three-phase study, we found high frequencies of 96G variants (where variant frequency is defined as the percentage of a specific haplotype or variant, in this case 96G, among all chromosomes in the population) and 96GG homozygous genotypes (91% and 83%, respectively) on the breeding farms. Over four years of selective breeding, these frequencies were markedly reduced, with just 37% of fawns carrying a 96G variant in the most recent fawning season and 7.8% identified as 96GG homozygous. Breeding outcomes followed expected patterns of Mendelian inheritance, with no evidence that any individual variant was deleterious in utero. On the two CWD-positive hunting properties, we found that CWD prevalence increased prior to our intervention, from approximately 61% to 79% on site 1 and from 4.3% to 59% on site 2. The overwhelming majority of the animals harvested and affected since the start of the study were homozygous for the 96G *PRNP* variant (cumulatively 300/390 of those harvested—77%, and 134/152 of those CWD-positive—88%). In the first year of controlled release and harvest of 19 animals on site 1, just two were found to be in early stages of CWD 12–16 months after their release; both were 96GG homozygous. Both breeding sites and the third hunting preserve remain CWD-negative. We plan to continue selective breeding into future years, ultimately eliminating the 96G variant from the herd, while continuously monitoring disease prevalence and other metrics on all properties.

## 2. Materials and Methods

### 2.1. Statement on the Humane Care and Use of Animals

Sample collection and genetic testing of the animals involved in this study were approved by the Animal Care and Use Committee of Midwestern University, #AZ-4603.

### 2.2. Background on Frequency Descriptions of PRNP Variants

Throughout the manuscript, we describe the frequency of various *PRNP* haplotypes or variants in whitetail deer. “Frequency” in these cases always refers to the percentage of a given haplotype or variant (e.g., 96G, 96S, 95H, 226K) among all chromosomes present in the population. Furthermore, it is important to note that animals carrying what is referred to as a 96G variant in the present manuscript have no other nonsynonymous SNPs at other locations. For example, while both the 95H haplotype (occasionally referred to elsewhere as H^95^G^96^A^116^Q^226^, or simply HGAQ) and the 226K haplotype (occasionally referred to elsewhere as Q^95^G^96^A^116^K^226^, or simply QGAK) have a guanine (G) at position 96, they are considered unique haplotypes and are not included in frequency calculations for either 96G (e.g., Q^95^G^96^A^116^Q^226^ or QGAQ) variants or 96GG genotypes.

### 2.3. Study Population

The study herd comprised several separated populations, consisting of two breeding sites with upward of 200 breeding does and a smaller number of breeding bucks, as well as three hunting preserves. All locations were fenced, isolating these populations from free-ranging deer and limiting both breeding and CWD exposure to those animals on-site. Breeding locations were typical of those found on whitetail deer farms and were made up of several pens housing ~10 deer each on cleared 2-acre plots with feed bunks and fresh water sources. Hunting site 1 includes approximately 360 acres of fenced property, composed of 270 acres of forested land and 90 acres of fields. Hunting site 2 is located on 480 acres of fenced wooded land. Hunting site 3 includes approximately 350 fenced acres of cedar swamps, planted pines, and agricultural fields. Prior to the beginning of the project, each of the hunting properties maintained 150–200 deer under conditions approximating those of free-ranging deer.

### 2.4. Amplification and Sequencing of the PRNP Gene

DNA was extracted from hair or archived semen samples from breeding animals and postmortem tissue samples from harvested animals, using a commercial extraction kit according to the manufacturer’s directions (Genejet Genomic Kit, Fisher Scientific, Hampton, NH, USA). At the start of the study, we used primers (223 and 224) and amplification conditions widely used in cervid *PRNP* genetic analyses, first published by O’Rourke and colleagues [33,37,38,39,42,43,44,45]. In the third and fourth year of the study, careful examination of the herd’s pedigree ultimately led to the identification of a polymorphism in the binding site for the forward (223) primer in an allele coding for a 96S variant (GenBank accession MZ773901). This polymorphism was responsible for a drastic reduction in amplification, resulting in sequencing challenges in animals carrying this specific allele. New primers were then developed, WTDPRNP-F (5′–TGT TTA TAG CTG ATG CCA CTG C–3′) and WTDPRNP-R (5′–ACA CCA CCA CTA CAG GGC–5′), which target the region just outside (e.g., 5′) of the original 223 forward and 224 reverse primer binding sites, respectively. Amplification conditions for this new primer set consisted of a 5 min, 95 °C hot start, followed by 35 cycles of 95 °C × 1 min, 55 °C × 1 min, and 72 °C × 1 min, and a final extension step of 72 °C for 5 min. Following amplification, products were run on a 1% agarose gel to confirm amplicon presence prior to sequencing (Genewiz, South Plainfield, NJ, USA). For cost and convenience purposes, sequencing was performed unidirectionally using reverse primer 224 in all years of the study, which provided a clear picture of single-nucleotide polymorphisms at *PRNP* positions 676, 347, 286, and 285, corresponding to haplotypes encoding 226K, 116G, 96G vs. 96S, and 95 variants, respectively. Genotypes were assigned to each animal on the basis of various nonsynonymous polymorphisms identified in the sequencing products.

### 2.5. Selective Breeding Program and Herd Management Strategy

A selective breeding program was developed in cooperation with the herd owner to incorporate two factors: (1) *PRNP* genotype, with 5–6 male whitetail deer with various *PRNP* genotypes selected for each breeding season, and (2) sires with subjectively desirable phenotypic traits, including antler phenotype and estimated Boone and Crocket score. Does were likewise selected on the basis of a combination of genotype and subjective and objective histories of producing fawns with desirable antler traits. The program designed was, thus, a careful balance of both desired genotypic and phenotypic outcomes and did not necessarily reflect the most efficient approach toward reducing susceptible variant frequencies.

On hunting premises 1, the herd management goals for the initial phase of the project were to steadily depopulate the native animals ranging on the property over the first 3 years (Figure 1). Once depopulated, animals of various genotypes would be released for progressively longer durations of time prior to harvest. In the initial release year (project year four), we planned to release 15–20 animals with various genotypes including those with highly susceptible 96GG genotypes and others carrying less susceptible 96S and 95H variants. On CWD-positive hunting premises 2 and CWD-negative hunting premises 3, our initial plan centered on a slow matriculation of animals with less susceptible variants onto the premises beginning in project year four. This would allow for their natural interbreeding with animals carrying the highly susceptible 96G variant, while harvesting a mix of released and native animals during regular hunting seasons.

During all 4 years of the first phase of the project, animals harvested in the field on each of the hunting premises were tested for CWD using conventional diagnostic testing, with our analyses focused solely on CWD-positive hunting premises 1 and 2. Initial CWD screening was conducted with immunohistochemical (IHC) evaluation of retropharyngeal lymph nodes (RLN) by the state veterinary diagnostic laboratory according to United States Department of Agriculture (USDA)-approved protocols [33,35,40]. Animals which were positive by preliminary IHC were subsequently confirmed by immunohistochemistry performed by the USDA’s National Veterinary Services Laboratory. Confirmatory IHC reports included both RLN and obex test results, providing very cursory information on disease stage; for example, cases which were positive in the RLN only were considered early in the course of infection, while those positive in both RLN and obex were considered in later stages of infection. 

### 2.6. Statistical Analyses

A comparison between expected and observed breeding outcomes was performed through a two-tailed chi-squared test with Yate’s correction. The analysis of genotypic prevalence and disease stage was likewise done through a conventional two-tailed chi-square test with Yate’s correction. Absolute *p*-values are presented where appropriate. All analyses were performed using Graphpad Prism 9 software.

## 3. Results

### 3.1. Dramatic Shifts in PRNP Variant Frequencies in Project Phase I

Prior to the breeding season in the first year of the study, *PRNP* genotypes of breeding animals were determined. Haplotypes encoding 96G (e.g., no other nonsynonymous SNPs) made up 91% of all total variants (Figure 2), with homozygous 96GG animals representing 83% of all breeding animals (Figure 3 and Table 1). Alleles coding for 95H, 96S, and 226K variants were much less common, making up roughly 0.39%, 5.7%, and 3.4%, of all variants, respectively. Genotypes of fawns born in the fawning season prior to the first year of the study were also determined, with gene frequencies and genotypes similar to that of breeding stock (Figure 2 and Figure 3 and Table 1). These values reflect the later identification of a poorly amplifying 96S haplotype identified in the third year of the project, discussed above, with reclassification of several adults and fawns.

With a clearer picture of gene frequencies and genotypes present in the herd, candidate breeding bucks were screened for the first year’s fall breeding season. Our initial goal was to simply reduce the frequency of the 96G variant, with males carrying 96GG, 96SS, 96G/226K, 95H/96G, and 95H/96S genotypes selected for breeding via artificial insemination or natural cover. Among fawns born in the first year of the study, gene frequencies showed considerable increases in the 96S and 95H variants, with modest increases in the 226K variant (12%, 7.0%, and 4.7%, respectively). In turn, fawns born homozygous for the 96G variant were also substantially lower than those born prior to the start of the study (56% in year 1 vs. 78% in the year prior to study initiation). These values likewise reflect the identification and reclassification of fawns carrying a poorly amplifying 96S allele discussed above.

On the basis of an incomplete picture of disease susceptibility provided by experiments in the field and in the lab [38,46,47], including studies evaluating similar polymorphisms thought to contribute to scrapie resistance in goats [21], we sought to increase the frequencies of the 95H and 226K variants in the second year of the study. Sires with 96G/96S, 95H/96G, 96G/226K, 96S/226K, and 95H/226K genotypes were selected for breeding in study year 2. The genotypes of fawns born in the second year of the study again showed sizable increases in the 95H and 226K variants (16% and 25%, respectively). The frequency of the 96S variants remained relatively unchanged from the first fawning season (12%), while the frequency of the 96G decreased steadily to 48%. Animals homozygous for 96G variants decreased remarkably between years 1 and 2, from 57% to 12%.

In the third and fourth years of the project, our focus shifted toward increasing the frequencies of the 95H and 96S variants as a result of a better understanding of CWD susceptibility provided by a large-scale field prevalence study in farmed deer [35]. In the third year, breeding was conducted using sires with 95H/96G, 95H/96S, 96G/226K, 96S/226K, and 95H/226K genotypes, while, in the fourth year, sires with 96G/96S, 96S/96S, 95H/96G, 95H/96S, 96S/226K, and 95H/226K genotypes were selected. Fawns born in the spring of the third and fourth years of the project continued to show drastic reductions in 96G variant frequencies, down to 46.0% in year 3 and 37% in year 4; 96GG homozygous animals continued a steady decline to 7.8% in in year 4, down from 12% in year 3. Concurrently, 96S variant frequencies increased to 20% in the third year of the study and 27% in the fourth year of the study. Animals carrying the 95H variant increased in frequency to 21% in year three and 23% in year four, while those with the 226K variant declined in the third year to 13.0%, with frequencies further reduced in year 4, at 12%. Year over year, variant frequencies and genotypes in fawns are presented in Figure 2 and Figure 3 and Table 1.

### 3.2. Selective Breeding for Less Susceptible PRNP Variants Results in Predictable Outcomes Based on Mendelian Inheritance

Over the course of first phase of the project, expected and observed breeding outcomes were recorded to determine whether they followed traditional Mendelian inheritance patterns. Total expected and observed numbers of each possible genotype across the four study years (Table 1) were not found to be significantly different (χ^2^ = 7.34, *df* = 9, *p* = 0.60), suggesting that none of the genotypes were inherently underrepresented in the offspring.

### 3.3. The 96GG Genotype Is Over-Represented in On-Site Cases of CWD

Over the first phase of the study, project years 1–4, samples collected for CWD testing purposes from all harvested animals on the two CWD endemic hunting premises were secondarily evaluated by *PRNP* gene sequencing. Results followed very closely with those of previous studies, with 96GG homozygous animals making up the vast majority of CWD cases on both premises (134/152, 88%). Odds ratios found CWD prevalence in 96GG deer to be 2.5 times that of 96G/96S animals (45% vs. 18%, respectively, *p* = 0.0004, Figure 4 and Table 2). Eighty-five percent of CWD-positive 96GG animals were in later disease stages (e.g., both obex- and RLN-positive, 115 of 135), compared to just 40% of CWD positive 96G/96S animals found in later disease stages (4 of 10, *p* = 0.0015, Figure 4 and Table 2). Prevalence and late disease stage values in 96G/226K animals were similar to those of 96GG animals (38%, *p* = 0.72 and 88%, *p* = 0.86, respectively), further supporting a focus on increasing 96S and 95H variants in fawns in the third and fourth years of the study. 

### 3.4. The Role of Selective Breeding in Reducing CWD Prevalence

As expected, CWD prevalence was found to increase on both CWD-positive hunting premises prior to our interventions. On hunting premises 1, CWD prevalence increased steadily from 61% (51 of 83 animals harvested) in year 1 to 62% (26/42) and 79% (11/14) in years 2 and 3, at which time all deer on the property had been depopulated and our management intervention began. In the fourth year of the project, 19 deer with a range of genotypes, including 15 96GG animals, two 96G/96S animals, and two with 95H/96G genotypes were released for 12–16 months prior to harvest. Following harvest, two of these deer (10.5%) were found to be CWD positive; both were 96GG homozygous and were positive in RLN tissue only (Figure 5 and Table 2).

Prevalence of CWD likewise increased on hunting premises 2 prior to our intervention, from 4.3% (three of 70 animals harvested) in year 1 to 24% (21/86) in year 2. In the third year of the study, prevalence continued to increase to 42% (18/43). Although our management plan for premises 2 and 3 initially sought to release animals carrying less-susceptible *PRNP* variants in the fourth year of the study, simultaneous with the harvest of animals born in the field, a decision was made to modify that plan and instead mirror our ongoing approach on premises 1 through a significant reduction in deer population numbers on sites 2 and 3 prior to introducing resistant animals. Without our planned release, prevalence on hunting site 2 climbed to 61% (20/33) in the fourth year (Figure 5 and Table 2), although it is important to note that all 20 animals testing CWD-positive from this site in year 4 were 96GG homozygous. Animals where CWD was not detected included four 96GG animals, seven 96G/96S animals, one 96SS animal, and one 95H/96G animal; both positive and negative animals had been born on-site and were approximately 3–4 years of age at the time of harvest.

To date, CWD has not been reported on the third hunting preserve or on either of the breeding locations.

## 4. Discussion

The management of chronic wasting disease in cervids has proven to be a challenging task for wildlife and agricultural agencies alike. With varying degrees of success, wildlife agencies typically rely on herd reduction and, in some cases, wide-scale depopulation efforts, among other regional management changes [48,49,50,51,52,53]. Federal and state agricultural agencies are traditionally tasked with overseeing CWD control efforts in farmed cervids, where positive herds are most often placed under quarantine and eventually depopulated [41]. Rarely, opportunities arise for managing the disease in privately owned herds without depopulation, using strategies tailored to the resources available to the individual operation. Previously, our group worked with a large hunting ranch raising North American elk (*Cervus canadensis*) in what was perhaps the first attempt to manage CWD in a private herd [28,54]. Although genotypic information was available, efforts to incorporate that information in our management objectives were unsuccessful. Instead, management focused solely on yearly live animal testing and removal of infected animals. Despite these measures, CWD prevalence increased steadily over a 3-year period, and the owners eventually elected to depopulate. Because of the relative difficulty in handling semi-domesticated whitetail deer compared to elk, live animal testing was not feasible in the present case. Our management efforts have instead focused on selective breeding and postmortem monitoring of disease prevalence among the various genotypic backgrounds.

In this preliminary phase of our study, tremendous progress was made toward reaching our primary objective: reducing the frequency of highly susceptible *PRNP* variants and increasing the frequency of less susceptible variants. This effort was guided by genotypic prevalence data from past studies and findings collected on-site in real time. Initial test results from our first year of release show promise in reducing CWD prevalence, with highly susceptible 96GG animals making up all identified infections. Subjectively, antler quality is as good or better than prior to intervention. In the second phase of our study, we continue to release animals with these variants for ever-increasing lengths of time to determine relative susceptibilities of variants that are exceedingly rare, at present, in either wild or farmed whitetail deer populations. Although the reasons for their relatively lower frequencies in many populations are unknown, we have not yet found any evidence to suggest that the 95H, 96S, or 226K variants are deleterious to the host. The third phase of the study will continue to follow subjective health and CWD prevalence in these variants, while monitoring any changes in several CWD-associated parameters, further discussed below.

With input from national and international wildlife and agricultural agency representatives, several potential concerns regarding our management efforts have been identified. It is important to note that, although our selective breeding efforts allow for a swift transition from highly susceptible to relatively resistant genotypes, shifts in *PRNP* variant frequencies are, in fact, happening at a much slower pace in free ranging cervids [55,56]. Many of the concerns expressed should, therefore, also apply to wild herds. As a result, our efforts in this captive herd may allow for more timely insight into what it means to be CWD-resistant, the risks of “silent carriers”, prion strain evolution, CWD diagnostic challenges, and alterations in disease pathology, pathogenesis, and zoonotic potential.

First, there is ongoing debate over what CWD resistance means in the context of cervid *PRNP* genotypes. It has been shown, unequivocally, that individual animals with several of the genotypes in the present study can be infected with CWD in either natural or experimental settings, including deer carrying 96SS and 95H/96S genotypes [35,37,39,57]. Often, these conditions represent the extremes of natural or experimental exposure: exceptionally high rates of prevalence in closed herds with high population densities [39] or arguably unnatural levels of oral or parenteral exposure in controlled experiments [37]; these conditions are likely atypical for free-ranging animals. At the same time, it has been repeatedly shown that both prevalence and disease stage are significantly limited in free-ranging and farmed whitetail deer carrying 95H, 96S, and 226K alleles, compared to highly susceptible 96GG homozygous deer [35,37,38,39,44,57,58]. It should be stressed that resistance in the context presented in the present study does not mean absolute immunity to infection; instead, it implies a measurably lower risk of infection and an altered disease course in animals with rare *PRNP* variants found to be infected. These differences between highly susceptible 96GG deer and those with resistant alleles, however, can present management windows that may be pursued in controlled environments through selective breeding. How effective these opportunities may be in free-ranging conditions or those approximating free-ranging conditions, and the disruptions that may arise in our current understanding of CWD biology and pathogenesis remains to be seen. Although we are not expecting to eliminate the disease from this herd entirely (we fully expect to find CWD-positive animals with less-susceptible *PNRP* genotypes going forward, especially as the frequency of these genotypes increases), we are hopeful that the trend toward lower and lower prevalence continues as we see an increasing frequency of less susceptible *PRNP* variants.

Second, the onset and the duration of CWD prion shedding in animals with rare alleles are poorly understood, and there is some concern that animals with less susceptible genotypes may serve as “silent carriers”, transmitting prions into the environment for longer periods of time [59]. Although the results of ongoing studies will better inform our understanding of disease transmission in these animals, the importance of shedding in a herd overly composed of resistant animals is understandably difficult to predict. High levels of resistant genotypes in a herd may parallel vaccination with incomplete protection, whereby a broad level of resistance within a herd has a measurable impact on disease prevalence and severity despite a basal level of shedding and transmission, as has been reported with SARS-CoV2, measles, and influenza [60,61,62,63]. Additionally, target harvest ages of both bucks and does could be adjusted in our controlled setting to permit the removal of animals prior to the estimated onset of infection or shedding. Increased transmission would be most evident in the third phase of the study, where we may see increasing rates of prevalence in animals with rare alleles; at this stage, harvest age optimization may better limit any likelihood of transmission.

A third concern is that CWD prion strains may adapt or evolve to infect animals with less common *PRNP* genotypes [64,65,66]. We acknowledge that this is a possibility, and the third phase of our study will monitor for this potential both in the field and in the laboratory. By assessing disease prevalence in various *PRNP* genotypes on-site over the second and third phases of the study, we may quickly identify shifting levels of susceptibility. In the laboratory, we may evaluate in vitro conversion capabilities of prions isolated from animals with rare *PRNP* genotypes and estimate CWD strain changes over time using the real-time quaking-induced conversion assay (RT-QuIC), for example [47]. Our selective breeding strategy would allow for relatively quick adjustments away from more susceptible alleles in favor of alternate *PRNP* variants, while providing for a nonhomogeneous *PRNP* background that may hinder the ability of CWD prions to adapt.

A fourth possibility is that CWD-infected animals with alternate *PRNP* variants may be more challenging to identify using conventional diagnostic tests, including ELISA and IHC. There is some evidence that prions derived from deer carrying 95H variants may be more sensitive to proteinase treatment [37], and that those isolated from deer with various *PRNP* genotypes may yield a diversity of Western blotting profiles that differ from that of the typical 96GG homozygous deer [66]. Working hand in hand with state and federal agencies, the second and third phases of the study will monitor for this potential and adapt accordingly, including modifying current testing protocols and incorporating newly developed detection assays, e.g., RT-QuIC, into diagnostic schema. 

Lastly, there is some concern that evolving CWD strains in cervids with rare *PRNP* genotypes may be more likely to be zoonotic than those strains currently identified [67,68,69,70,71,72]. It is important to note that the converse may be just as likely, i.e., any new CWD strains recovered may prove less infectious in various human model systems. As with previous concerns, we will address this possibility in the second and third phases of the study with a combination of in vivo and in vitro investigations, including collaborations with groups experienced with animal model systems such as humanized transgenic mice [73,74,75] and nonhuman primates [69,70], as well as RT-QuIC evaluation of any newly isolated strains in human PrP substrate [76]. Importantly, identifying any variations in zoonotic risk in a controlled environment would allow us to better predict the outcome of and prepare for the eventual genotypic shifts due to selective pressure reported in wild cervid populations highlighted above.

In summary, although this study represents the first of its kind to use selective breeding to manage chronic wasting disease in farmed cervids, our strategy is not without precedent, and we acknowledge the decades of work done in both sheep and goats that have led to international declines in scrapie prevalence. There is much about CWD resistance that is known, including correlations of prevalence and disease stage with specific *PRNP* genotypes; however, much remains to be discovered, including pathogenesis and transmission kinetics in rare genotypes, strain evolution and adaptation, and zoonotic potential. Our results from this initial phase of the project are very promising, with dramatic changes in *PRNP* variant frequencies and an encouraging first year of harvest prevalence data. As the study evolves, project goals are likely to change based on new findings on disease prevalence, CWD strains, or CWD resistance, for example, through genome-wide analysis studies [77], leading to more robust strategies for selective breeding in the future. It is our genuine hope that our study is continually improved upon, and that the data collected may benefit both farmed and wild cervids through the eventual containment and eradication of this devastating disease.

## Figures and Tables

**Figure 1 genes-12-01396-f001:**
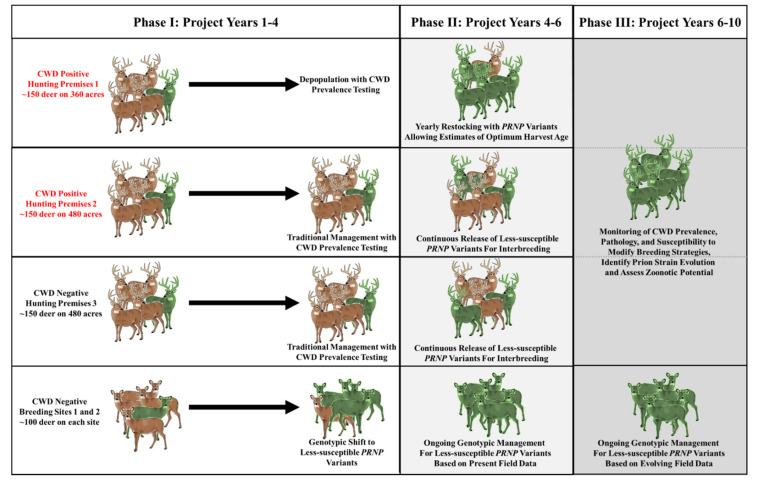
**Project overview**. This schematic highlights phase I strategies for hunting premises 1, 2, and 3, initially planning for a “clean slate” approach on premises 1—a gradual depopulation followed by systematic introduction of animals with various genotypes, including those with resistant genotypes (shaded green). On hunting premises 2 and 3, a more gradual introduction of animals with resistant genotypes was initially planned, with genotypic shifts expected in situ as these animals bred with animals of more susceptible genotypes. This plan on premises 2 and 3 was later modified to be more in line with the clean slate strategy for premises 1, based on rising prevalence rates on site 2. On both breeding sites, selective breeding and genotypic shifts would be balanced with desirable phenotypic traits, eventually leading to complete removal of 96G *PRNP* variants (e.g., those animals without nonsynonymous SNPs at other positions) in favor of 95H, 96S, and 226K variants. These animals would be used to stock each of the hunting premises. In phases II and III, data on prevalence, pathology, transmission, and strain evolution will be used to modify selective breeding strategies as necessary. CWD, chronic wasting disease.

**Figure 2 genes-12-01396-f002:**
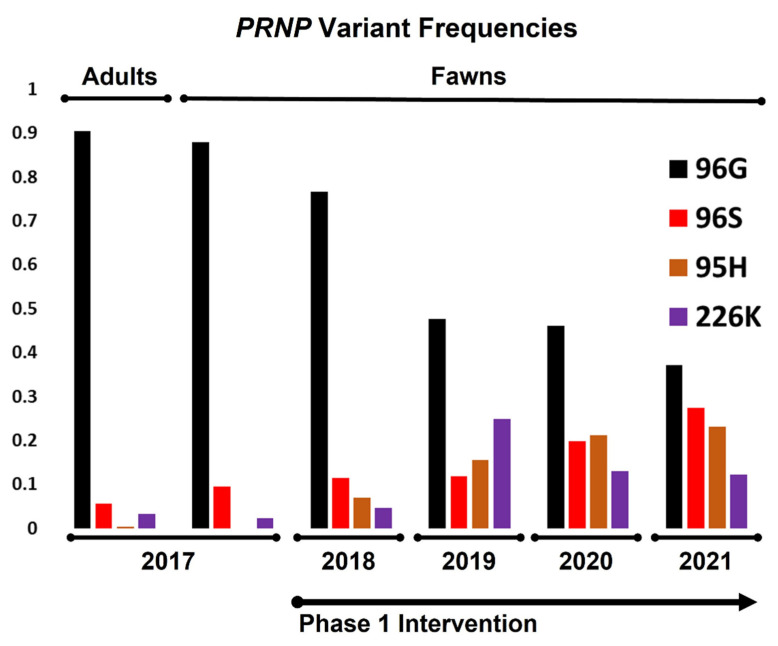
**Changes in PRNP variant frequencies among breeding animals and fawns, both prior to and following selective breeding intervention.** The frequency of 96G variants, e.g., the percentage of 96G variants (those variants without non-synonymous SNPs at other positions) among all chromosomes in the population, shifted markedly from a high of over 90% to less than 40% over the first phase of the project, with concurrent gains seen in 95H, 96S, and 226K variants.

**Figure 3 genes-12-01396-f003:**
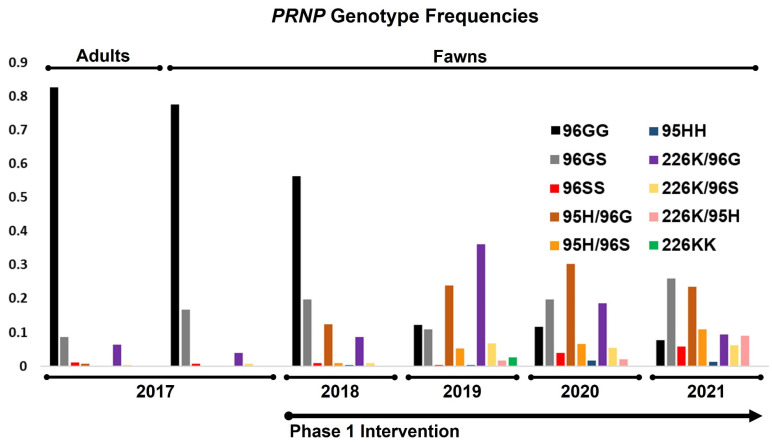
**Changes in PRNP genotype frequencies among breeding animals and fawns, both prior to and following selective breeding intervention.** The frequency of 96GG genotypes shifted markedly from a high of over 80% to less than 10% over the first phase of the project, with concurrent gains seen in 95H, 96S, and 226K carriers.

**Figure 4 genes-12-01396-f004:**
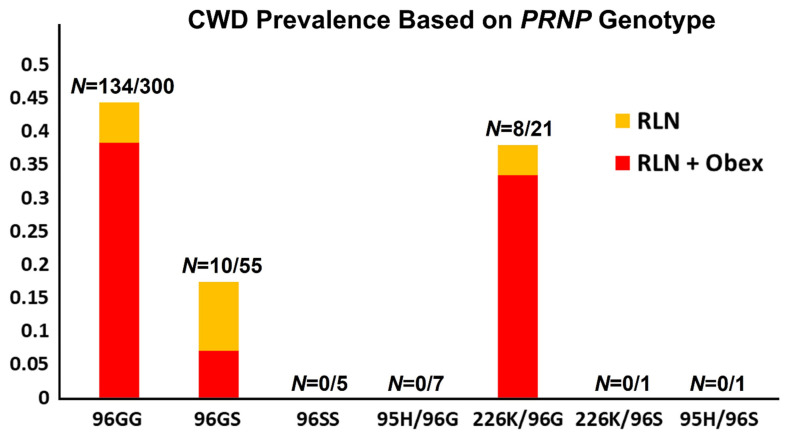
**Prevalence of chronic wasting disease on hunting premises 1 and 2 based on genotype.** As in previous studies, the 96GG genotype was overly represented among CWD-positive animals, two and a half times that of 96GS animals. Affected 96GG animals were also twice as likely to be in advanced stages of disease. Prevalence and disease stages of animals with 96G/226K genotypes were similar to those observed with 96GG animals. Only those genotypes found in harvested animals, which include animals from the first phase of the project and the initial year of phase two, are shown. Data from hunting premises 3, which has to date remained CWD-negative, are not included in the figure.

**Figure 5 genes-12-01396-f005:**
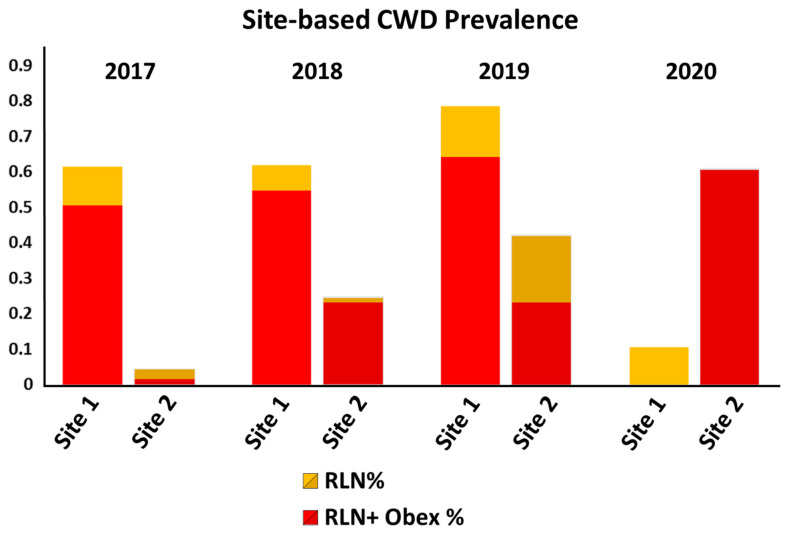
**Prevalence of CWD on hunting sites 1 and 2 over the course of the study.** Prevalence steadily increased on site 1 over the first phase of the project, followed by an abrupt drop with the initiation of study phase 2 (2020) and the introduction of animals with resistant *PRNP* genotypes. Prevalence likewise increased on site 2 through the initiation of study phase 2, when our initial plans for in situ genotypic shifts were modified to allow for complete depopulation of the site prior to introducing animals with resistant genotypes, mirroring objectives for site 1. Data from hunting premises 3, which has to date remained CWD-negative, are not included in the figure.

**Table 1 genes-12-01396-t001:** ***PRNP* genotypes present in adult breeding animals and fawns throughout the course of the study.** Data include genotypes present prior to selective breeding efforts (2017), as well as expected and observed numbers of fawns in the initial phase of the project. Expected and observed numbers were statistically similar (two-tailed chi-square test with Yate’s correction), indicating predictable patterns of Mendelian inheritance for the *PRNP* gene.

Group	Year	Exp./Obs.	96GG	96GS	96SS	95H/96G	95H/96S	95HH	96G/226K	96S/226K	95H/226K	226KK
**Adults**	**2017**		427	45	6	4	0	0	33	2	0	0
**Fawns**	**2017**		97	21	1	0	0	0	5	1	0	0
	**2018**	**Expected**	128.75	52.25	8	28	3	0.25	19.25	1.75	0	1.75
		**Observed**	137	48	2	30	2	1	21	2	0	0
	**2019**	**Expected**	45	26	5.25	73	12.5	1	117.25	19.25	5.5	5.25
		**Observed**	38	34	1	74	16	1	112	21	5	8
	**2020**	**Expected**	50.75	74.5	17	104	24	3.75	55.5	16	6	1.5
		**Observed**	41	70	14	107	23	6	66	19	7	0
	**2021**	**Expected**	30.75	97.75	23	90.25	39.25	7.5	31.75	27.25	25.5	1
		**Observed**	29	97	22	88	41	5	35	23	34	0

**Table 2 genes-12-01396-t002:** **CWD prevalence among different *PRNP* genotypes on hunting sites 1 and 2 over the course of the study.** Results include information from animals which were in early disease stages (positive in the retropharyngeal lymph nodes, RLN, only), as well as those in later disease stages that were positive in both the RLN and obex region of the brainstem. Animals with the 96GG genotype were overly represented among CWD-positive cases and tended to be in more advanced stages of disease compared to animals with the 96GS genotype. Data from hunting premises 3, which has to date remained CWD-negative, are not included in the table.

Project Year	Hunting Site	CWD Status	96GG	96GS	96SS	95H/96G	95H/96S	96G/226K	96S/226K	Totals
**2017**–**2018**	**Site 1**	Negative	25	5	0	0	0	2	0	32
		RLN Only	6	3	0	0	0	0	0	9
		RLN + Obex	39	1	0	0	0	2	0	42
	**Site 2**	Negative	53	10	1	1	0	2	0	67
		RLN Only	1	0	0	0	0	1	0	2
		RLN + Obex	1	0	0	0	0	0	0	1
**2018**–**2019**	**Site 1**	Negative	10	1	0	0	0	4	1	16
		RLN Only	2	1	0	0	0	0	0	3
		RLN + Obex	19	1	0	0	0	3	0	23
	**Site 2**	Negative	42	14	2	3	1	3	0	65
		RLN Only	1	0	0	0	0	0	0	1
		RLN + Obex	19	1	0	0	0	0	0	20
**2019**–**2020**	**Site 1**	Negative	2	1	0	0	0	0	0	3
		RLN Only	2	0	0	0	0	0	0	2
		RLN + Obex	6	1	0	0	0	2	0	9
	**Site 2**	Negative	17	5	1	0	0	2	0	25
		RLN Only	6	2	0	0	0	0	0	8
		RLN + Obex	10	0	0	0	0	0	0	10
**2020**–**2021**	**Site 1**	Negative	13	2	0	2	0	0	0	17
		RLN Only	2	0	0	0	0	0	0	2
		RLN + Obex	0	0	0	0	0	0	0	0
	**Site 2**	Negative	4	7	1	1	0	0	0	13
		RLN Only	0	0	0	0	0	0	0	0
		RLN + Obex	20	0	0	0	0	0	0	20
**Cumulative Totals**		Negative	166	45	5	7	1	13	1	238
		RLN Only	20	6	0	0	0	1	0	27
		RLN + Obex	114	4	0	0	0	7	0	125

## Data Availability

All applicable data are included in the tables in this manuscript.

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
