# Peer review of "Selective Breeding for Disease-Resistant PRNP Variants to Manage Chronic Wasting Disease in Farmed Whitetail Deer"

_genes, 2021, doi:10.3390/genes12091396_

Round 1

Reviewer 1 Report

Chronic wasting diseases is fast spreading across deer in North America. In white-tailed deer, the prion gene PRNP encodes the prion protein PrP. Three main variants of PrP have been reported. The two less common ones have synonymous differences at codon 95 and 96. These two PrP proteins are substantially less common among CWD-positive deer than among CWD-negative deer, indicating a strong protective effect against CWD.

The study is an excellent contribution to the literature of genetic effects on CWD, and is furthermore well conducted and very well written. It examines shifts in protective SNP frequencies after selective breeding on a white-tailed deer ranch, and cases of CWD among animals with different SNPs.

Most importantly, it reports a flaw in previously developed primer, that is likely to be extremely relevant to genetic studies of CWD, and also developed alternative better primers that will be welcome by those conducting studies.

The main issue is one of nomenclature, as described below. Essentially, it needs to be clearer that “frequencies” refers to total chromosomes (2N) in a population. And it needs to be much clearer that 96G does not refer to variants encoding 96G but rather excludes those that do encode 96G but that also have a different mutation elsewhere (e.g., 95H haplotypes also have 96G, but are not designated as “96G”).

-Perhaps a paragraph on nomenclature is justified at the beginning of methods, stating that the 96G designation excludes 96G deer that may have non-synonymous variation at other sites, such as 95S, 226K etc.

-Likewise, the phrase “The 96G designation is given only to haplotypes without non-synonymous SNPs at other positions” should be used liberally in figure and table legends and across the manuscript, to emphasize this.

Additional thoughts:

-Up to the authors, but would the new primers be worth indicating in the abstract, if there is room? “In the present study” is not needed, and “With the use of novel primers” could precede “we found that breeding…”

-Figure 1 was cut off on the right in my pdf

-Figure 1, perhaps clarify what is meant by “premises” also should the singular “premise” be used in some cases?

-Figure 2 legend: please make clear that frequency here is a reference to “all chromosomes in the population”.

-Table 1 “statistically similar” perhaps add in parentheses what test was used.

-Table 2 “not included in this **table**” not figure

-From line 127 on, it is important that all number/frequencies indicate whether they are referring to deer or to chromosomes. For example, if 50% of the chromosomes are 96G, then 25% of the deer will have two copies of 96G and 50% will have one copy. Thus the frequency of 96G could be listed as 50% (at the chromosome level) or 75% (at the deer level). Thus please specify whether frequency is referring to “chromosomes in the population” or “deer (with one or two copies of 96G)”—though alternative wording is fine. In many cases it is clear what is being referred to, but in some cases less so.

31-southern, lower case

44-keep DNA and protein conceptually separate. Here the phrase beginning “--specifically” might be replaced by “, and the encoded primary structure of the PrP protein(s),” or something similar. (I don’t like referring first to DNA and then “specifically” to the protein.) But if the authors find it fine, then they can keep it as is.

75-the phrase “these polymorphisms are mutually exclusive” seems a bit too strong, perhaps add the qualifier “so far”. Also since 226K is shown in the figures, please somewhere in the paper indicate its relationship to the other SNPs (is 96G encoded in the 226K haplotype)

76-allele is used, but this often refers to a single SNP so perhaps “haplotype” is better. Or at least define allele in an initial paragraph on nomenclature in the methods.

151-needs to specify the degree to which the deer in the study sites were isolated from other wild deer, in terms of mating and/or potential exposure to CWD

179-I think the primer names certainly need to be changed. Currently they use “PrP” which is the protein name. Primer names should never incorporate the protein name, since primers don’t amplify proteins. Perhaps use “PRNP” instead. While this may cause some within-lab difficulty, the long-term consequences of using a protein name rather than a target gene name are worse.

180-“bound” should be “bind” or even better: “target”

188-“This approach gave…” this sentence is very difficult to understand. Please rewrite.

208-“those with”

249-“Alleles coding for 96G made up 91% of all total variants” Do you mean “haplotypes encoding 96G (but not other non-synonymous SNPs) were carried by 91% of chromosomes” or something similar? Also, not clear whether 96G here means all chromosomes with 96G or just those not carrying other variants?

-For haplotypes encoding 226K, do they also encode 96G? Then someplace early on the nomenclature needs to be clearly defined that 96G does not refer to 96G but to deer that have 96G and don’t have 226K. The same needs to be stated for 95S. If the 95S/96G deer are not being counted as 96G, then this all needs to be specified early on, and the wording needs to be well thought out in the abstract and elsewhere.

-Likewise the difference between haplotype and allele needs to be considered. While not consistent in the literature, haplotype usually refers to all the sites across the gene, while allele sometimes is used for an individual SNP. Thus in line 250, “Alleles coding for 96G made up…” might be written “Haplotypes coding for 96G (but not other non-synonymous SNPs) made up 91% of total chromosomes”

258-“poorly amplifying 96S allele” might better be called “haplotype” since haplotypes and not SNPs are amplified.

398-“seldom successful…” may be too negative. Perhaps “With varying degrees of success…”

426-“their rarity in nature” seems too strong, as these alleles are not very uncommon in some populations. Perhaps “their relatively lower frequencies is unknown”

480-“even a likelihood” seems too strong, as this is not known, perhaps exclude this phrase?

Discussion: when referring to positive results, it is not always clear whether the 226K animals are being distinguished in the positive results from the 96G animals (that are not 226K).

Author Response

Please see the attachment for our responses to Reviewer #1.

Reviewer 2 Report

<Major points>

Resistant alleles against CWD have been identified in cervids. For examples, less susceptible cervid alleles include the 96S and 95H allele in white-tailed deer. On the basis of the background, this study developed a selective breeding program for farmed white-tailed deer in order to manage the future project to reduce CWD prevalence.

The strategies and the activities as well as the limitations of this study are totally well-written, and the manuscript contains interesting data. There are, however, some minor points that need to be fixed.

<Minor points>

(1) Line 226-240: Methods used for CWD testing are not clearly described. Preliminary IHC evaluation by USDA approved protocols as well as the confirmatory IHC should be concreted by a more detailed description or should be referred to some papers.

(2) Line 242-246: Software used for statistical analysis should be included.

(3)“PRNP” should be italicized in the title/abstract as well as the main texts.

Author Response

Please see the attachment for our response to Reviewer #2.
